# GASA Proteins: Review of Their Functions in Plant Environmental Stress Tolerance

**DOI:** 10.3390/plants12102045

**Published:** 2023-05-21

**Authors:** Mohamed Taieb Bouteraa, Walid Ben Romdhane, Narjes Baazaoui, Mohammad Y. Alfaifi, Yosra Chouaibi, Bouthaina Ben Akacha, Anis Ben Hsouna, Miroslava Kačániová, Sanja Ćavar Zeljković, Stefania Garzoli, Rania Ben Saad

**Affiliations:** 1Biotechnology and Plant Improvement Laboratory, Center of Biotechnology of Sfax, B.P “1177”, Sfax 3018, Tunisia; 2Faculty of Sciences of Bizerte UR13ES47, University of Carthage, BP W, Bizerte 7021, Tunisia; 3Plant Production Department, College of Food and Agricultural Sciences, King Saud University, P.O. Box 2460, Riyadh 11451, Saudi Arabia; 4Biology Department, College of Sciences and Arts Muhayil Assir, King Khalid University, Abha 61421, Saudi Arabia; 5Biology Department, Faculty of Science, King Khalid University, Abha 9004, Saudi Arabia; 6Department of Environmental Sciences and Nutrition, Higher Institute of Applied Sciences and Technology of Mahdia, University of Monastir, Mahdia 5100, Tunisia; 7Institute of Horticulture, Faculty of Horticulture, Slovak University of Agriculture, Tr. A. Hlinku 2, 949 76 Nitra, Slovakia; 8Department of Bioenergy, Food Technology and Microbiology, Institute of Food Technology and Nutrition, University of Rzeszow, 4 Zelwerowicza St, 35601 Rzeszow, Poland; 9Centre of the Region Haná for Biotechnological and Agricultural Research, Department of Genetic Resources for Vegetables, Medicinal and Special Plants, Crop Research Institute, Šlechtitelů 29, 77900 Olomouc, Czech Republic; 10Czech Advanced Technology and Research Institute, Palacky University, Šlechtitelů 27, 77900 Olomouc, Czech Republic; 11Department of Chemistry and Technologies of Drug, Sapienza University, P.le Aldo Moro 5, 00185 Rome, Italy

**Keywords:** GASA protein, phytohormonal responses, stress tolerance, plant development, redox regulation

## Abstract

Gibberellic acid-stimulated *Arabidopsis* (*GASA*) gene family is a class of functional cysteine-rich proteins characterized by an N-terminal signal peptide and a C-terminal-conserved GASA domain with 12 invariant cysteine (Cys) residues. GASA proteins are widely distributed among plant species, and the majority of them are involved in the signal transmission of plant hormones, the regulation of plant development and growth, and the responses to different environmental constraints. To date, their action mechanisms are not completely elucidated. This review reports an overview of the diversity, structure, and subcellular localization of GASA proteins, their involvement in hormone crosstalk and redox regulation during development, and plant responses to abiotic and biotic stresses. Knowledge of this complex regulation can be a contribution to promoting multiple abiotic stress tolerance with potential agricultural applications through the engineering of genes encoding GASA proteins and the production of transgenic plants.

## 1. Introduction

Plant growth is a reaction to external stimuli and internal signals processed through a complex network that produces a response and growth pattern peculiar to a particular species. Our understanding of the signaling cascades initiated by adverse environmental phenomena has increased owing to the identification of several gene families in the model plant *Arabidopsis thaliana* L., where GASA (Gibberellic acid-stimulated *Arabidopsis*) is one of those proteins (also referred to as Snakin protein) with an array of cysteine-rich (Cys) peptides regulated by gibberellins (GAs) [1,2,3,4,5,6]. In recent years, the number of studies that have focused on a cysteine-rich protein that plays a role in the development and biotic and abiotic stress tolerance in a wide range of plants has increased [7,8,9,10,11,12,13,14,15,16,17]. To date, GASA peptides were identified in diverse monocotyledonous and dicotyledonous plant species [18]. The first member of the GA-stimulated transcript (*GAST1*) gene was identified in a tomato and observed to be upregulated by GA_3_ treatment in the GA-deficient (*gib1*) mutant gene [19]. The *GASA*s encode low-molecular-weight peptides, which are characterized by an N-terminal signal peptide and precisely conserved 12-Cys C-terminal, which is induced by hormones and involved in signaling pathways that modulate hormonal interaction and endogenous levels of the hormone [18,20,21,22]. However, knockout or missing key peptide residues (Cys) may cause the generation of a non-functional GASA domain, causing the protein’s loss of its biological functions [6,22]. In fact, the Cys-rich regions remained relatively constant throughout evolution, thereby contributing significantly to plant biological processes [23].

To provide a focused overview of the *GASA* gene family, this review aims to elucidate their mechanisms of action and putative functions, including their participation in hormone crosstalk and redox homeostasis during plant responses to biotic or abiotic stress. Through a collection of information on the *GASA* gene family’s structure, subcellular localization, and biological function, a better understanding of their molecular properties and versatile biochemistry can be gained, enabling more efficient and conscious biotechnological applications. Notably, there is compelling evidence that GASA proteins are involved in complex crosstalk that occurs between various hormones at the levels of biosynthesis and action. Although most research has focused on the connections between these genes and GA responses, increasing evidence suggests that *GASA* genes interact with other hormones. To explain the role of GASA proteins in stress tolerance and developmental processes, this review proposes a simplified model, highlighting their involvement in redox and hormone homeostasis (Figure 1). To fully comprehend the complex networks of GASA proteins in plants, future research should aim to identify cell surface receptors and downstream components of GASA signaling. Furthermore, investigating the relationship between hormone homeostasis and redox status may provide further insights into the mechanisms underlying GASA protein function.

## 2. GASA Proteins Identified in Plants

GASA proteins, belonging to a multigene family, are found in a large number of plants. However, they are not found in animals. Nonetheless, homologous genes are found in some bacteria, such as *Escherichia coli*, *Klebsiella pneumoniae*, *Nitriliruptoraceae bacterium*, *Acinetobacter baumannii*, *Soehngenia saccharolytica*, *Glycocaulis profundi*, and *Staphylococcus warner*. Whether or not these genes code for GASA peptides requires future investigation. Silverstein et al. [24] identified approximately 445 genes encoding GASA proteins in 33 plant species using comprehensive genome sequence analysis. Moreover, results of bioinformatics data mining indicate that *GASA* genes are localized in several species of vascular plants. However, they are missing in moss and green algae, suggesting that the emergence of *GASA* genes may have been an adaptation of ancestral plants to dry land [25]. Many *GASA* gene families have been identified using whole-genome analyses of different plant species [26]. For example, *Zea mays* and *Orysa sativa* possess 10 *ZmGASA* and 10 *OsGASR* members, respectively, and 37 members of the *GmGASA* family were identified in *Glycine max* [27,28,29]. In addition, 37 *TaGASA* genes in the common wheat genome (*Triticum aestivum*, hexaploid) were characterized [14], and 19 *TdGASA* genes are known in *Triticum durum* [17].

GASAs proteins have received growing attention in studies of plant-pathogen resistance, stress tolerance, and growth regulation owing to their significance as key growth regulators and antimicrobial agents in some non-food cash crops. For instance, poplar, an important tree species, also used for vehicle seat belts, contains 21 candidate genes *GASAs* in *Populus trichocarpa* and 19 genes in *Populus euphratica* [30]. In addition, some members play a significant role in drought-stress responses, vegetative organ growth and development, and hormone responses [30,31]. For example, 26 *MdGASA* genes are present in the apple (*Malus domestica*) genome, some of which are involved in flower induction [22]. Tobacco (*Nicotiana tabacum*) has 18 *NtGASA* genes, and some of them exhibited distinctive patterns of expression in various tissues, suggesting their potential involvement in tobacco plant development [32]. Some genes in the 18-member *CcGASA* family respond to *Xanthomonas citri* infection, the primary cause of fruit loss in *Citrus clementina* [33]. *Canavalia rosea*, a perennial climbing herb, has 23 putative *CrGASA* genes involved in numerous physiological and biological processes, displaying complex and diverse functions [16]. Finally, some of the 14 *VvGASA* genes identified in grapes (*Vitis vinifera*) may be involved during different phases of seed development in seedless grape cultivars, and in other tissues [34].

Analysis of the members of the *GASA* gene family in several plant species revealed that GASA proteins exhibit significant diversity in the number of family members. Thus, the GASA peptide length varies significantly (ranging from 31 to 1172 aa) in the aforementioned plants (Table 1). In particular, the gene sequence of the *GASA* members in common wheat is substantially longer (>261 aa) than in other plant species [35]. Moreover, the number of family members (between 1 and 40) also varies widely between different plant species. This considerable variety contributed to the lack of a common subfamily definition and indicates the need to structurally and functionally characterize the *GASA* genes to better understand their biological relevance, despite the remarkable sequence similarity by subheadings. It should provide a concise and precise description of the experimental results, their interpretation, as well as the conclusions that can be drawn.

## 3. GASA Protein’s Structure

GASA proteins are typically small (~7 kDa) and positively charged, while Cys-rich proteins [54] are involved in plant defense responses, such as antimicrobial activity against a wide range of phytopathogens [55,56,57,58,59,60,61] and animal pathogens [62,63], as well as in a variety of plant-development processes [6,27,38,46,64]. Almost all *GASA* genes encode a low molecular weight protein comprising three domains: (1) An N-terminal 18-to-29 amino acid signal peptide; (2) A hydrophilic middle segment that differs greatly in length and amino acid composition within members of the same family and across species; and (3) A highly conserved C-terminal GASA domain made of around 60 amino acids with 12 Cys residues that contribute to the protein’s stability [2,65]. The prediction of the three-dimensional structure shows a uniform flexible arrangement with mainly coils and α-helices, and little-to-no β-sheets [17,22,34,43]. This is supported by the X-ray results showing the helix-turn-helix motif stabilized by disulfide bonds [66]. The conservation of the GASA domain can be observed in many studies where all GASA proteins share a common motif [X_n_CX_3_CX_2_RCX_8(9)_CX_3_CX_2_CCX_2_CXCVPXGX_2_GNX_3_CPCYX_10(14)_KCP] (where X can be any of the 20 proteinogenic amino acid residues except for cysteine) [67]. While the stability of the GASA domain is assured by the disulfide bonds between cysteine residues, it is known that members of the same GASA family form a different model of bonds, such as the case of Snakin-1 and Snakin-2 in potato where CysXI and CysIX bonds with CysVIII and CysX, respectively, in Snakin-1 [62], while in Snakin-2 they bond with CysV and CysI, respectively [68]. Previous studies have demonstrated that the mode of formation of cysteine disulfide bonds directly affects the spatial protein structure and function [69]. GASA proteins typically form five-to-six disulfide bonds, which may be necessary for the senior structure of GASA and crucial for GASA protein interactions with other proteins [3,46]. GASA protein is involved in the regulation of plant growth and development through interacting with different proteins, such as rice *OsGSR1*, which is involved in the regulation of GA and BR crosstalk through an interaction with brassinosteroid (BR) synthase DIM/DWF1 protein [5]; potato Snakin-1 proteins can interact with itself to play a role in the regulation of cell division in the potato leaf [20]. Among the various classes of plant antimicrobial peptides, the GASA family peptides are the most cysteine-rich compared to the other large families of the plant host defense peptides (HDPs), which have less than 10 cysteine residues [70]. Crystallography and HPLC-ESI-QTOF analyses reveal that up to six disulfide bonds are formed by the 12 highly conserved cysteine residues [71]. A helix-turn-helix (HTH) motif that is common to GASA peptides is deciphered from the 3D structure by X-ray and mass spectrometry data [66,72]. According to these findings, the spatial structure of GASA is dependent on the disulfide bonds and the HTH motif, which may also be essential for the GASAs interactions with its target (such as the cell membrane, protein, and DNA).

## 4. Subcellular Localization of GASA Proteins

A series of experiments have attempted to determine where the GASA proteins are located in planta, as the protein’s function is largely related to its subcellular location. Although the function of a GASA protein is related to its GASA domain, its subcellular location is linked to its signal peptide. This implies that such a protein may belong to a class of secreted proteins. Bouterra et al. [17] showed by performing in-silico predictions that the majority of durum wheat TdGASA proteins are estimated to be extracellular, except for two, i.e., TdGASA4 and TdGASA12 that seem to be located in the membrane and endomembrane, respectively. Rice OsGASR-GFP fusion proteins were briefly expressed in onion epidermal cells, indicating that either OsGASR1 or OsGASR2 are primarily localized in the cell wall or apoplast [5,64]. In *Arabidopsis*, AtGASA4 and AtGASA6 are located at the cell periphery if they contain a signal peptide. However, they are found in the nucleus if the signal peptide is missing [4]. In addition to the aforementioned subcellular distribution, the non-cleaving signal peptide AtGASA4 has been hypothesized to attach to the endoplasmic reticulum (ER) [2]. The location of only a few GASA proteins is determined experimentally. The immunoblot analyses suggest that GIP1 (*Petunia hybrida*’s GASA) is localized in the ER membrane. Just like GIP1, GIP4 is potentially localized in the ER membrane. However, based on their cleavable hydrophobic N-terminal segment, GIP2 and GIP5 are estimated to be localized in the cell wall [46]. In addition, the rubber tree HbGASA5 and HbGASA9 proteins are found all over the cytoplasm and nucleus [72]. The citrus CcGASA4::GFP fusion protein emits a fluorescent signal in the nucleus and plasma membrane [73]. *Nicotiana benthamiana* agro-infiltrated leaves contain potato StSN1—GFP fusion proteins, which are located throughout the plasma membrane [20]. Importantly, the transient expression of StSN1 protein in insect cells reveals that the subcellular location of the peptide is restricted to the cytoplasm. However, its mature form is located in the nucleus, even though StSN1 does not have a potential nuclear localization signal (NLSs) [56]. Collectively, each member of the GASA family has a different subcellular localization (cell wall, cytoplasm, plasma membrane, nucleus, and ER), and their shift from the cell periphery to the nucleus may be crucial to their antimicrobial function. However, more detailed research in this respect needs to be done to gain insight into the biological role of GASA proteins. The subcellular localization of GASA proteins studied experimentally in plants was summarized (Table 2).

## 5. Tissue and Organs Specific Expression Patterns of *GASA* Genes

The expression of *GASA* genes was determined in different tissues and organs in many studies. Overall, these studies demonstrated that *GASA* genes are differentially expressed in different organs regarding spatial and temporal regulation. For example, in potato, *Snakin-1* transcripts are found to be widely present in axillary, stem, floral buds, and fully developed petals. However, transcripts were not found in roots, stolon, or leaves [61], while its proteins are present primarily in young tissues with active growth and cell-division zones [75]. Moreover, *Snakin-1* promoter’s activity declines over time as the plant ages [76]. *Snakin-2* is developmentally expressed in tubers, stems, flowers, shoot apex, and leaves, but not in roots [57]. Unlike *Snakin-1* and *Snakin-2*, the third potato Snakin (*Snakin-3*) is expressed in roots, stems, and axillary buds [21]. In rice’s developing panicles, *OsGASR1* was only found in florets with high levels, whereas *OsGASR2* was found in both florets and branches, but it was not present in mature and flag leaves. This indicates that *OsGASR1* and *OsGASR2* are important in meristems and the development of panicles [64]. *OsGASR9* transcripts were detected in high levels in panicles, while in leaves, they were low [74]. In an apple’s leaves, buds, and flowers, high levels of expression of *GASA* genes were observed. However, three of these genes (*MdGASA3*, *MdGASA13*, and *MdGASA26*) are more active in fruits [22]. In strawberries, *FaGAST1* is mainly expressed in ripe fruits and roots [38]. In durum wheat, high transcripts levels of *TdGASA2, TdGASA5*, *TdGASA9*, *TdGASA14*, *TdGASA18*, and *TdGASA19* were revealed in leaves, whereas *TdGASA17* was accumulated in stems. The *TdGASA6*, *TdGASA11*, and *TdGASA15* genes showed significantly greater expression in roots than in other tissues. Although the expression levels differ across tissues, *TdGASA3* showed a high expression in all tissues [17]. Just like *TdGASA3*, *MsSN1* expression was found in all tissues studied in alfalfa, including roots, stems, leaves, and immature floral buds [25]. In *Peltophorum dubium*’s developing seedlings, *PdSN1* transcript levels are 40 times higher than in adult leaves [72]. The transcript of *VvGASA1* and *VvGASA2* genes showed high levels in grapevine leaves, while *VvGASA9* and *VvGASA10* genes were high in fruits and seeds [34]. In *Hevea brasiliensis* tree, *snakin-1* was accumulated in the early stages of leaf development [77]. In *Arabidopsis*, the *AtGASA14* gene expression was detected in young leaves and the elongation zone of roots [6]. The *AtGASA4* promoter-directed GUS has strongly stained in the vegetative shoot apical meristems and imitating leaves [78]. However, the promoter of the *AtGASA5* gene is active in the root hairs, the basal portion of the roots, the shoot apex, and the inflorescent meristems [79]. The in-situ hybridization study in maize demonstrates that *ZmGSL2*, *4*, *6*, and *9* are expressed in newly formed lateral root primordia, indicating that *GASA* genes play a role in lateral root development [27]. Additionally, the *GASA* genes are most active at specific developmental stages. In petunia, *GIP2* was expressed in the elongating stem and corollas, whereas *GIP4* and *GIP5* are expressed at earlier development stages [46,80]. Furthermore, the *SN1* promoter has been shown to be active in the early stages of the plant and gradually decreases as the plant ages [76]. Moreover, Zhang et al. [16] studied the expression profiles of *CrGASA*s genes family of *C. rosea* and reported that *CrGASA*s are expressed at higher levels in the flowers or fruit than in the leaves, vines, and roots.

Overall, these findings support that the *GASA* genes are tissue/developmental stage-specific expressed. The majority of them are highly expressed in young plant tissues and organs, in the vigorous growth site, or in reproductive and storage organs, indicating their role in plant growth and development as well as in the first line of defense.

## 6. Involvement of *GASA* Genes in Plant Growth and Development

Members of the *GASA* family are involved in numerous physiological processes in plants where members of the same *GASA* family may have similar, different, or opposite functions. Interestingly, they all seem to be associated with young tissues and organs that are actively growing, which suggests that *GASA* genes are engaged in biological activities like cell division or expansion. Indeed, the simultaneous suppression of *AtGASA4* and *AtGASA6* leads to late blooming in *Arabidopsis*. However, early flowering is achieved by an overexpression of *AtGASA6* [81]. Silencing *AtGASA4* in transgenic plants results in a reduction of seed weight and yield, as well as an abnormal shoot and flower phenotype, while overexpressing it leads to an increase in seed weight and yield. [78]. The overexpression of *AtGASA5* in *Arabidopsis* reduces stem growth rate and delays flowering *AtGASA5* delays the onset of flowering by increasing the expression of flowering repressor FLC (Lowering Locus C) and decreasing the transcript levels of the flowering factors FT (Flowering Locus T) and LFY (Leaf Y) [79]. Additionally, silencing potato *St-GSL1* (*StSN1*) altered the cell-wall composition of leaves, influenced cell division and metabolism, and led to a smaller size of leaves [20]. In gerbera, *GEG* and *PRGL* have opposite functions, while *PRGL* increases the size of petals by promoting cell elongation [41,82], but *GEG* limits the size of petals by inhibiting cell elongation [40]. Petunia *GIP4* and *GIP5* play a role in cell division, whereas *GIP1*, *GIP2*, and tomato *GAST1* seem to display a similar ability to enhance stem elongation by promoting cell elongation [19,46]. The transcript of *ZmGSL2*, *4*, *6*, and *9* genes are detected in emerging lateral root primordia in maize, revealing their function in lateral root development [27]. In strawberry, the ectopic overexpression of *FaGAST1* resulted in a delay of the plant and a reduction in the size of fruits caused by an inhibition of the cell elongation during fruit development [38]. In contrast, by influencing cell expansion in spikelet hulls, *OsGASR9* positively regulates maize grain size and yields [74]. Similarly, the two haplotypes (H1 and H2) of *TuGASR7* in wheat (*Triticum urartu*) exhibited pleiotropic effects on grain weight and yield [83]. In grapevine, *VvGASA5* was highly expressed in seedless fruits but undetectable in seeded fruits, which suggests its involvement in ovule abortion [34]. Furthermore, based on omics data analyses, *GASA* genes were highly induced in young tissues during proliferation and contributed to the panicle differentiation process with a gain in grain size/weight [27,39,64,84,85]. The apple *GASA* genes are highly induced during the seedling and flower stage [22].

*CcGASA4*, of *Citrus*, was shown to inhibit the synthesis of lignin and flavonoids and may play a role in plant–pathogen interactions, inhibiting growth, and affecting flowering development [33].

Interestingly, *GASA* genes may be involved in cellular processes like cell division or expansion, as they all correlate with young tissues and actively growing organs.

## 7. Phytohormones and GASA Proteins

Phytohormones are frequently recognized as growth regulators, but they also play a significant role in response to various stresses both biotic and abiotic. Gibberellic acid (GA_3_) participates in various plant developmental processes, including flowering [86], cell-division promotion [87], and stem elongation [88]. The *GASA* family genes have been reported to be modulated by GA_3_, abscisic acid (ABA), and other phytohormones [67]. It has been shown that numerous *GASA* genes are responsive to exogenous GA_3_ treatment, and the transcript level of six out of 15 *AtGASA* genes (*AtGASA4/6/7/8/13*) are shown to have increased transcript levels after GA_3_ treatment [4,89] and all of the four Petunia *GIP* (*GIP1/2/4/5*) genes [46]. Just like the aforementioned *GASA* genes, maize *ZmGSL1/2/4/6/9* [27], rice *OsGASR1/2* and *OsGSR1* [5,64], *Salvia miltiorrhiza SmGASA4* [49], *Fagus sylvatica FsGASA4* [37], *HbGASA4-2/7-1/13/14-1/16* [72], apple *MdGASA1/6/7/19* [22], and all of durum wheat *GASA* genes [17] are upregulated by exogenous GA_3_. However, GA does not similarly affect all *GASA* gene expressions. *Arabidopsis AtGASA1/5/9/11* [4], potato *snakin-2* [57], and apple *MdGASA13/26* [22] are downregulated by GA_3_. *GASA* genes’ response to exogenous GA_3_ also seem to be different depending on the tissue or the development stage, such as in the case of soybean *GsGASA1*, which is upregulated in leaves but downregulated in roots [40]. *Arabidopsis AtGASA4* is repressed in roots and leaves but promoted in meristem tissues [2] and apple *MdGASA5*, which is promoted initially but becomes inhibited during the flower-induction period [22]. Although the name *GASA* is derived from their responsiveness to GA_3_, some genes are unaffected by it like *AtGASA2/3/10/12/14/15* and *SN1* gene [4,61].

Alongside GA_3_, GASA proteins are also responsive to exogenous ABA. Indeed, ABA enhances *AtGASA2/3/5/14* expression but reduces the expression of *AtGASA7/9* in *Arabidopsis* [4] and all *TdGASA* genes except for *TdGASA10* in durum wheat [17]; it also promotes the expression of *snakin-2* in potato but reduces that of *snakin-3*, while *snakin-1* is not affected by ABA [21,57]. In apples, all *MdGASA* expressions are promoted by ABA [22], and in *Hevea brasiliensis*, *HbGASA4-2/14-1/15* are highly responsive to exogenous ABA [72]. *GASA* genes can also be affected antagonistically by ABA and GA, such as in the case of *snakin-2* in potato [57], *FsGASA4* in *Fagus sylvatica* [37], *GIP1* in petunia [80], *GsGASA*1 in *Glycine soja* [42], and *GAST1* in tomatos [19]. *GASA* genes can also be induced by other phytohormones. In *Hevea brasiliensis*, *HbGASA7-1/14-1/14-3/15/16* is upregulated by jasmonic acid (JA), while *HbGASA4-2/14-1/15/7-1* is slightly responsive to salicylic Acid (SA) where *HbGASA15/7-1* shows continuous expression after 24 h of treatment [72]. While *GASA* gene expression is either promoted or inhibited by phytohormones, it can in turn influence the level of some phytohormones. While *OsGARS1* expression is promoted by GA, it activates a BR biosynthetic enzyme to regulate BR levels, which in turn inhibits its expression [5]. The knockdown of the *snakin-1* gene in potato plants induced reactive oxygen species (ROS), SA, and GA accumulation, and ABA and sterol biosynthesis decrease [75]. Moreover, the overexpression of *FsGASA4* promotes the expression of SA biosynthesis-related genes, which positively impacts the levels of endogenous SA [37]. Furthermore, *AtGASA6* was integrated with the GA, ABA, and glucose-signaling pathways to achieve the regulation of seed germination by promoting cell elongation [90].

Collectively, these results proved that GASA proteins maintain redox balance and play essential roles in complex crosstalk among several hormonal signaling pathways. Therefore, they are involved in the development and plant responses to environmental stressors.

## 8. Involvement of GASA in Abiotic Stress Tolerance and Redox Status Homeostasis

Numerous studies support that *GASA* genes are highly implicated in abiotic stress responses [6,68,91]. For instance, the expression of *OsGASR1* in rice is induced by UV irradiation [64]. The expression of *TdGASA1* in durum wheat was increased following the exposure of durum wheat seedlings to salt and osmotic treatment [17]. Indeed, the yeast cells overexpressing the *TdGASA1* gene exhibited enhanced growth compared to the control cells under NaCl, mannitol, LiCl, H_2_O_2_, and heat treatments [17]. The expression of *CrGASA*s showed habitat- and environmental-stress-regulated patterns in *C. rosea* and the heterologous induced expression of some *CrGASA*s in yeast-enhanced tolerance to H_2_O_2_. In addition, some *CrGASA*s showed elevated heat tolerance and heavy metal (HM) cadmium/copper (Cd/Cu) tolerance. Other reports support that *AtGASA14* is positively regulated by abiotic stresses [6], while the *AtGASA5* gene expression is negatively affected by abiotic stress, especially heat stress [88]. In the past few years, Zhang et al. [35] proved that wheat *TaGASA1* (homolog of rice *OsGASA1* gene) was involved in heat-stress responses. In rice, the highest *OsGASA1* transcript level was induced by salt treatments [91]. Another rice GASA family member, *OsGASA3*, also showed the highest transcript level under salt stress [85]. Ko et al. [92] reported that *Arabidopsis AtGASA4* overexpression enhances heat-stress tolerance. Sun et al. [6] showed that *AtGASA14* is involved in abiotic-stress responses via modulation of reactive oxygen species (ROS) accumulation. Other reports suggest that the heat-induced expression of *AtGASA5* gene promotes SA signaling regulation and heat-shock-protein accumulation [93]. Similarly, the *Fagus sylvatica FsGASA4* gene expression in *Arabidopsis* lines confers tolerance to salt, oxidative, and heat stresses [94]. Zhang et al. [35] demonstrated that *TaGASA1* from wheat, which is the homolog of *OsGASA1*, was regulated by heat stress. Indeed, in rice, the highest transcript level of rice *OsGASA1* was regulated by salt and ABA stresses [91]. Similarly, *OsGASA3* from rice showed a high accumulation of transcript by salt stress [84]. However, *OsGASA* family genes in responses to biotic stress have not been well studied so far. Recently, in the cacao plant, *TcGASA1/14* transcript levels were shown to be induced under osmotic and salt stress, while the *TcGASA16/17* transcript showed the opposite result [53]. *GsGASA1* isolating from *Glycine soja* was reported to be involved in inhibiting root growth via the accumulation of DELLA proteins under chronic cold stress [42]. The DELLA protein GAI is upregulated in *GASA5*-overexpressing plants, and *GASA5* is downregulated by exogenous GA_3_. Thus, GA signaling may mediate the degradation of the GAI protein through *GASA5* [79]. On the other hand, in *Phaseolus vulgaris*, *Pvul-GASA-18* expression is enhanced under salt stress in all tissues. However, *Pvul-GASA-1* was induced in leaves and inhibited in roots, while *Pvul-GASA-16/23* levels were not affected in roots but decreased in leaves under the same conditions of salt stress [47]. In *Canavalia rosea*, all *CrGASA* genes were promoted in leaves after 2 h of Mn treatment and after 48 h in roots. *CrGASA* also showed an increase in transcript levels after long-term treatment of Zn and Cu. However, *CrGASA3/5/10/1* was initially promoted but then inhibited after 48 h of Cd treatment in both leaves and roots. Under salt stress, *CrGASA2/14/16* showed a temporary peak. However, only *CrGASA7/16* was responsive to heat stress [16]. In contrast, all *TdGASA* genes seem to be responsive to salt and osmotic stress, and the overexpression of *TdGASA* genes in yeast showed better growth compared to non-transgenic control yeast under salt, osmotic, LiCl, H_2_O_2_, and heat stress [17]. Similar results were observed in five *CrGASA* genes in which overexpression enhanced H_2_O_2_ resistance in transgenic yeast and complemented Cd sensitivity, while there were no obvious effects on yeast cells treated with Zn, Co, Ni, or Mn [16]. Additionally, the *GASA* gene from tobacco is expressed in yeast alleviated Cd toxicity [95]. In the same way, the *SmGASA4* gene isolated from *S. miltiorrhiza* was involved in secondary metabolites biosynthesis and its ectopic expression in *Arabidopsis* lines promotes their development and abiotic stress tolerance [49]. Overall, GASA proteins may participate in hormone homeostasis as integrators of internal and environmental cues, adjusting the balance of cell growth promotion and inhibition to regulate plant development and stress tolerance.

Although ROS serves as a secondary messenger in many important physiological processes, such as cellular signaling transmission, it can also cause oxidative damage when exposed to a variety of environmental stressors, such as salinity and heavy metals [96]. The GASA peptides have been characterized as hormonal-signaling transducers/integrators, directly linked to the biosynthesis and transduction processes [3]. Considering that they have putative redox-active sites (cysteine residues), the *GASA* family members can influence ROS accumulation [3,6,23,89]. This can be observed in the reduction of ROS levels in *Arabidopsis* overexpressing *AtGASA4* by repressing the accumulation of H_2_O_2_ and nitric oxide in wounded leaves [89]. In addition to its capability to enhance salt and heat tolerance in plants overexpressing *FsGASA4*, it also enhances oxidative stress tolerance [37]. While H_2_O_2_ promotes *GIP2/4/5* expression, the overexpression of *GIP2* in petunia prevents the accumulation of H_2_O_2_ caused by osmotic stress or ABA in stomatal guard cells and wounded leaves [3].

Additionally, higher levels of ROS were detected in *SN1* silencing lines, indicating an altered redox balance [20]. Furthermore, scavengers of ROS, such as ascorbate, galactinol, and raffinose, were significantly reduced in *SN1*-silencing lines compared to wild-type plants. This supports the hypothesized role for this molecule in regulating cell division, further suggesting that *SN1* may perform its action by regulating ascorbate accumulation [20]. The GASA domain is closely related to protein function. *AtGASA4* inhibits ROS accumulation and provides partial resistance to NO donor sodium nitroprusside (SNP) in seeds. Indeed, *E. coli* expressing either the whole *AtGASA4* or a shortened *AtGASA4* with only its GASA domain intact, was resistant to SNP, but mutant *GASA4* lost its redox activity and capacity to trigger GA responses when conserved cysteines were replaced by Alanine [89]. In summary, GASA proteins probably execute their physiological function via its conserved cysteine-rich domain through the redox and hormonal signaling pathway. 

## 9. Involvement of GASA Proteins in Biotic Stress 

Like hormones and environmental stress, *GASA* genes are also induced by biotic stress like viruses, fungi, and bacteria, which suggests their involvement in the plant’s innate immune system [67]. GASA is the major component of the HDPs. Potato Snakin-1 (StSN1) [61] and Snakin-2 (*StSN2*) [57] are the first peptides isolated from this family which mediate strong antimicrobial activity against plant pathogens [55,57,61]. The role assigned to *GASA* genes was based on their expression profile [57,61,97,98] and the effect of their expression in transgenic plants [55,60,99,100]. Overexpression of *snakin-1* or *2* in transgenic potato and tomato plants enhanced tolerance to *Rhizoctonia solani* and *Erwinia carotovora* [55,99,100], while its expression in transgenic wheat showed significant improvement in vivo against the take-all disease caused by *Gaeumannomyces graminis var. tritici* [60]. However, silencing *snakin-2* gene in *Nicotiana benthamiana*, induced susceptibility to *Clavibactermi chiganensis* [101]. In addition, the knockdown of potato *snakin-1* gene affects cell wall composition, cell division, and leaf primary metabolism, suggesting that *StSN1* may also be involved in numerous cellular processes [20]. In vitro *StSN1* and *StSN2* play a large spectrum of anti-bacterial and anti-fungal bioactivity [57,61,71]. Consistent with *StSN1*, *MsSN1*, and *Snakin-1* from alfalfa and *Solanum chacoense*, respectively, it has proven in-vitro and in-vivo antimicrobial activity against several bacterial and fungal pathogens [25,50]. Nahirñak et al. [21] showed that potato *Snakin-3* expression was linked to the plant defense process, as its gene expression levels were significantly induced upon pathogen infection. Intriguingly, *HbGASA* genes from the *rubber tree* were regulated by *C. gloeosporioides* (fungal pathogens) where *HbGASA7-1/15/16* transcript levels increased while *HbGASA6* transcript levels decreased after one day of inoculation [72]. Similarly, *TcGAS*A was highly expressed in tolerant Cacao after 24 h of *P. megakarya* inoculation [53]. Additionally, citrus *CcGASA4* transcript levels increased significantly in leaves after an infection with *Citrus tristeza virus* [73]. Moreover, transgenic wheat constitutively expressed *Solanum chacoense snakin-1*, which had a reduced and delayed infection with *Blumeria graminisf.* sp. *Tritici* [50]. *StSN1* and *StSN2* expression under the control of the potato light-inducible *Lhca3* promoter led to enhanced resistance against blackleg disease with no morphological penalties in potato plants [100]. Boonpa et al. [102] suggest that *OsGASR3* from rice-mediated defense against pathogens and plant development in growing seedlings and during panicle formation.

Interestingly, the GASA family genes protect plants from viruses and nematodes besides their antifungal and antibacterial activity. Additionally, *turnip mosaic virus* and *Soybean mosaic virus* resistance was enhanced in *GmSN1* transgenic *Arabidopsis* and soybean [103]. After an infection with the *Citrus tristezavirus*, the citrus homolog gene *CcGASA4* is highly induced in citrus leaves [73]. Pepper *CaSn* was confirmed to be activated by nematode infections and plays a major role in its host defense [36].

Intriguingly, GASA proteins are attractive biotechnological targets due to their biological properties, particularly for the creation of novel agents for disease control [54]. Heterologous expression is a widely used method for medium- and large-scale production. To produce many proteins on a large scale and at a low cost in hosts like bacteria, yeast, fungi, and plants, some systems (producing organism/vector) were developed [104,105,106]. In particular, recombinant Snakins has been produced with success in *E. coli*, *Pichia pastoris*, and, more recently, in baculovirus/insect cells [56], which exhibit expected antibacterial and antifungal properties [36,58,59,71]. Recently, a synthetic Snakin-1 made from the natural form of potato *StSN1* has been shown to have a significant inhibitory effect against some food spoilage yeasts, but it does not pose a risk to human consumption [107]. 

This suggests that the GASA peptides could be used to protect food, pharmaceuticals, or cosmetics from microorganisms’ decomposition.

## 10. Conclusions

Crop-yield losses are primarily caused by environmental constraints. This situation is compounded further by climate change, the limited availability of land, and the rapid spread of devastating diseases. It is urgent to cultivate plants that can better withstand both biotic and abiotic stresses to achieve food security. This will necessitate a thorough understanding of the various mechanisms that control stress tolerance. Several studies demonstrated the central role played by different families of GASA proteins during biotic and abiotic stresses. In the meantime, they play active roles in numerous aspects of plants’ growth and development process. They have great biotechnological potential in the fields of pathology and agronomy due to their outstanding properties. Despite the progress in our understanding of the significant function that the genes of *GASA* family play in plants, there are still a few unanswered questions. To investigate some of the GASA peptides’ more significant and specific characteristics, further in-depth structural and functional confirmations are required. On the other hand, these proteins have been shown to function as antioxidants, but the nature of this antioxidant activity and whether it has a direct or indirect effect on physiological responses and development are still unknown. Certainly, protein–protein interaction analysis would better clarify the function of this family. Finally, it requires further in-depth studies to uncover the underlying mechanism(s) and the contributions of these valuable proteins before their future agricultural applications.

## Figures and Tables

**Figure 1 plants-12-02045-f001:**
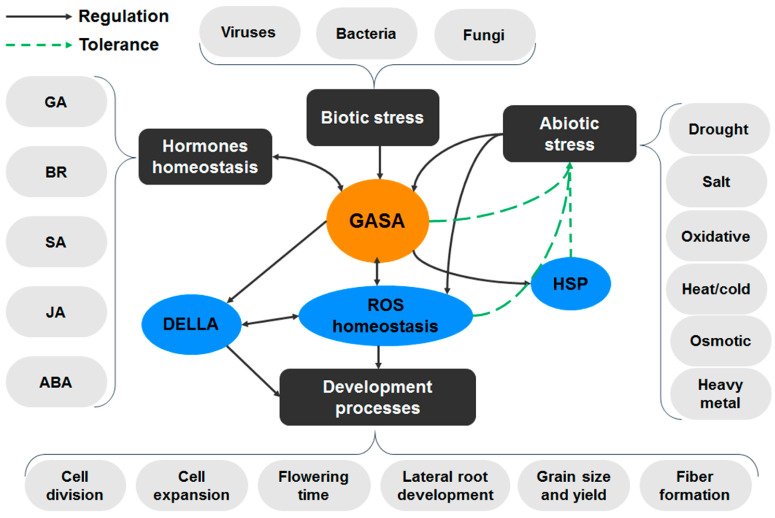
Hypothetical model of GASA proteins roles. *GASA* gene expression is modulated in response to different hormones, biotic, and abiotic stress. Through their conserved cysteine-rich domain, the encoded proteins maintain redox homeostasis, and participate in developmental processes and stress tolerance. The involvement of *GASA* genes in plant growth and stress responses was mediated by DELLA and heat-shock protein (HSP). However, the functional connection between these different components is still unknown.

**Table 1 plants-12-02045-t001:** Overview of the characterized GASA protein family in plant species.

Species	Protein Family	Characterized Members	Protein Length (aa)	Ref.
*Arabidopsis thaliana*	AtGASA	15	87–275	[22]
*Arachis duranensis*	AdGASA	20	Ø	[31]
*Arachis ipaensis*	AiGASA	22	Ø	[31]
*Arachis hypogaea*	AhGASA	40	60–202	[31]
*Benincasa hispida*	ø	9	80–232	[15]
*Canavalia rosea*	CrGASA	23	70–233	[16]
*Capsicum annuum*	CaSn	1	104	[36]
*Citrullus lanatus*	ø	9	88–216	[15]
*Citrus clementina*	CcGASA	18	70–206	[33]
*Cucumis melo*	ø	10	80–222	[15]
*Cucumis sativus*	ø	9	61–231	[15]
*Cucurbita moschata*	ø	10	85–516	[15]
*Fagus sylvatica*	FsGASA	1	107	[37]
*Fragaria × ananassa* *(strawberry)*	FaGAST	2	86–91	[38,39]
*Gerbera hybrida*	GEG PRGL	1 1	150 (PRGL)	[40,41]
*Glycine soja*	GsGASA	1	97	[42]
*Glycine max*	GmGASA	37	66–198	[28]
*Gossypium arboreum*	GmGASA	17	69–213	[43]
*Gossypium barbadense*	GbGASA	33	73–870	[43]
*Gossypium herbaceum*	GheGASA	19	89–926	[43]
*Gossypium hirsutum*	GhGASA	38	76–264	[43]
*Gossypium raimondii*	GrGASA	25	72–297	[43]
*Hevea brasiliensis*	HbGASA	16	88–275	[44]
*Lagenaria siceraria*	ø	8	80–212	[15]
*Luffa cylindrica*	ø	9	85–221	[15]
*Malus domestica*	MdGASA	26	88–305	[22]
*Medicago sativa*	MsSN1	1	108	[25]
*Momordica charantia*	ø	15	59–370	[15]
*Nicotiana tabacum*	NtGASA	18	61–147	[32]
*Oryza sativa*	OsGASA	10	92–152	[29]
*Peltophorum dubium*	PdSN	12	63–95	[45]
*Petunia hybrida*	GIP	4	Ø	[46]
*Phaseolus vulgaris*	PvulGASA	23	88–179	[47]
*Phyllostachys edulis*	PheGAST	8	81–113	[48]
*Populus euphratica*	PeuGASA	19	88–222	[30]
*Populus trichocarpa*	PtrGASA	21	88–191	[30]
*Salvia miltiorrhiza*	SmGASA	1	110	[49]
*Sechium edule*	ø	16	86–223	[15]
*Solanum chacoense*	Snakin-1	1	Ø	[50]
*Solanum lycopersicum*	SlGASA	17	88–146	[51]
*Solanum tuberosum*	Snakin	18	88–143	[21]
*Sorghum bicolor*	SbSN	12	93–137	[52]
*Theobroma cacao*	TcGASA	17	88–320	[53]
*Trichosanthes anguina*	ø	18	80–234	[15]
*Triticum aestivum*	TaGASR	37	261–1172	[14]
*Triticum durum*	TdGASA	19	92–222	[17]
*Vitis vinifera*	VvGASA	14	74–298	[34]
*Zea mays*	ZmGSL	10	75–128	[27]

**Table 2 plants-12-02045-t002:** GASA proteins subcellular localization.

GASA Protein	Subcellular Localization	Signal Peptide Length (aa)	Ref.
AtGASA14	Plasma Membrane	Ø	[6]
CcGASA4	Cell Membrane—Nucleus	Ø	[73]
GIP	Plasma Membrane—Endoplasmic Reticulum	19 (GIP1)	[46]
GsGASA1	Plasma Membrane—Cytoplasm—Nucleus	27	[42]
HbGASA5—HbGASA9	Nucleus—Cytoplasm	Ø	[72]
OsGASR1—OsGASR9	Apoplasm—Cell Wall	29	[64,74]
PRGL	Cell Wall	19	[41]
SlGASA1	Cytoplasm—Nucleus	18–29	[51]
Snakin-1	Plasma Membrane	25	[20,21]
TaGASR1	Cell Membrane—Cytosol	Ø	[35]

## Data Availability

Not applicable.

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
