# Peer review of "GASA Proteins: Review of Their Functions in Plant Environmental Stress Tolerance"

_plants, 2023, doi:10.3390/plants12102045_

Round 1

Reviewer 1 Report

Its a nice written review. I believe after some editions, its going to be great! I have some changes to be addressed In word, attached.

The english is ok, well written.

Author Response

Reviewer #1:

-I believe "insight" is more of a term where one generates new hypothesis etc... I think I would change the word "insight" for “Review of the functions… etc. (optional)

Response: Thanks for your suggestion. The title  has been changed as recommended.

-line61-62, mention which aa may have a loss-of-function. Also, how do you know its not a gain of function. Please clarify if its known or not.

Response: Thanks for your suggestion. The key amino acids are the conserved Cys residues. The mutated Cys residues result in non-functional GASA protein. The sentence has been revised to be clear for the reader.

-Line 77-82: Please re-phrase, not sure what you want to say here. 

Response: Thanks for your suggestion. As recommended, the paragraph was re-phrased to be clear for the reader.

 -In table 1 it is not correct to state that the number of members is that fixed number that is included in the table when it is "at least" that number, especially in those publications that do not make a complete study of the family in that species but only a particular member. Also, mention if the full genome is assembled, if so, check there for GASA annotations, not only papers.

Response:  We agree with reviewer’s comments, thank you for pointing this out. Almost all the plants genomes mentioned in Table 1 have been assembled and their predited genes have been in silico annotated. Therefore, to make the revised Table 1 more rational, we have mentioned the number of the characterized GASA full length members in each plant species.

-The described snakin-z member (57) is actually a partial sequence, so you should clarify it in the table.

Response: Thanks for your suggestion. We have removed the partial Snakin-z from the revised Table 1.

-Plant host defense peptides 432 (HDPs) must be defined the first time it appears in the text (line 166)

Response: Thanks for your suggestion. The definition has been moved to the first time HDPs are mentionned.

-Line 174. Protein function is related "principally" to its subcellular location. Not sure I agree with principally. Its important yes. Please rephrase.

Response: Thanks for your suggestion. We changed “principally” with “largely” to make the meaning clearer.        

-Line 204 : The word "on" should be remove.

Response: Thanks for your suggestion. The word have been removed from the text.

-Line 257.  In transgenic plants, you mean over-expression or silencing? Please explain.

Response: Thanks for your suggestion. AtGASA4 gene silencing and overexpressing have different effects. The sentence was changed to highlight both biotechnological approaches.

-line 300. Define Cd Cu in text.

Response: Thanks for your suggestion. Both elements have been defined.

-Line 310-311: Check the agreement between what the sentence states and the reference.

Response: Thanks for your suggestion. This sentence has been removed.

-Line 312. Remove the word "Recently" since it refers to a text from 2017

Response: Thanks for your suggestion. The word “Recently” have been replaced with “In the past few years”.

-Subheading 8. Since phytohormones also modulate the expression of the GASA protein, it is confusing because they are excluded in subtitle 7, perhaps referring to "crosstalk" is more precise.

Response: Thanks for your suggestion. This part has been restructured in the revised version of the manuscript and the subheading 7 has been integrated into the other subheadings.

-Is there a subfamily definition for the different members of snakins? If so Include.  If not, explain there is no subdivision due to nucleotide differences.

Response: Thanks for your suggestion. Some studies reported phylogenetic trees that devided GASA proteins members into 3 to 4 groups based on their peptide sequences and their conserved motifs, but no common subfamily definition for the different snakins members have been reported. As recommended, we have mentioned that no common subdivision has been reported due to nucleotide sequence variabilities.

-Figure 2. Please re do model. Maybe make a model to explain better. Does not add much.

Response: Thanks for your suggestion. This figure has been removed.

-Line 399-402: Add the reference to the work of Nahirñak et al., (Plant Signaling & Behavior 7:8, 1-5; August 2012) who had written something very similar.

Response: Thanks for your suggestion. The reference has been added as recommended.

-Line434: Remove the comma after "this family" because it contradicts Line 56 (GAST1) since they were not the first members of all GASAs but the first to be described as antimicrobial.

Response: Thanks for your suggestion. The comma has been removed from the text.

-Line 449-451: Snakin-z was a partial protein of only 31 aa that does not have the complete GASA domain. Please, check if its inclusion in this work is correct.

Response: Thank you for pointing this out. Although the partial Snakin-z shares antimicrobial activity with GASA proteins, we have removed this part from the revised manuscript.

-Line 467: Write "Pepper CaSn was" … instead of "Pepper CaSnis"

-Line 494: Remove the "-" before the word "On"

Response: Thanks for your suggestion. We have corrected these remarks as recommended.

Reviewer 2 Report

Several studies demonstrated the central role played by different families of GASA proteins during biotic and abiotic stresses; in the meantime, they play active roles in numerous aspects of plants growth and development process.

Despite the progress in our understanding of the significant function that the genes of GASA family play in plants, there are still few unanswered questions, and there is no extensive review on the topic, yet. Data available only on dedicated plant families or species e.g. Populus or lettuce or cotton. So this review can be a gap filling study.

List of comments:

1. Please correct: in the title "Gasa", elsewehere GASA.

2. In line 56-57: "The first member of the GA-stimulated transcript (GAST1) gene was identified in tomato and characterized as GA-deficient (gib1) mutant gene." Please revise this sentence, this statement is not correct.

3. In line 68-69: " ...subcellular localization, biological function, and their participation in hormone crosstalk  and redox homeostasis" please complete with "of GASA proteins"

4. Line 66-82: here the authors tried to highlight the aim of the review, please rewrite it, to be more focused, as repetions are occuring, and cause diffuse sturture.

5. Please distinguish more clearly when you mention information about GASA proteins or GASA genes in all over the text. E.g:

In line 183:

"In Arabidopsis, AtGASA4 and AtGASA6 are located 183 at the cell periphery if they contain a signal peptide, however, they are found in the nu- 184 cleus if the signal peptide is missing [4]. In addition to the aforementioned subcellular  distribution, AtGASA4 with non-cleavable signal peptide was speculated to attach to the  endoplasmic reticulum (ER) [2]." 

The genes are not indicated in italics, however in the reference studies genes were presented. Title of one of them:"Expression patterns of GASA genes in Arabidopsis thaliana: the GASA4 gene is up-regulated by gibberellins in meristematic regions"

Please try to make these parts more focused.

Or as 4. subsection is about protein localisation, remove all the information about genes, to the next section about "5. ...GASA genes".

6. Again, it is not logical that in section 7.  Modulation of GASA proteins’ expression:   "The expression of GASA genes was determined in different studies. Overall, these studies showed that GASA genes are differentially expressed in different organs or tissues  and under different treatments (Figure 2)." Why here again the localisation of genes are presented in a separeted section, and not involved in the section 5. Please try to follow a more logical structure and/or suitable subsection titles need.

7. Figure 2 is not enough informative.

8. From line 305 the relationship between GASA and hormones are mentioned, however the next section is about "Phytohormones and GASA proteins" So, please consider a focused arrangement.

Author Response

Reviewer #2:

  1. Please correct: in the title "Gasa", elsewehere GASA.

Response: Thanks for your suggestion. “GASA” has been capitalised as recommended.

  1. In line 56-57: "The first member of the GA-stimulated transcript (GAST1) gene was identified in tomato and characterized as GA-deficient (gib1) mutant gene." Please revise this sentence, this statement is not correct.

Response: Thank you for pointing. This statement has been revised.

  1. In line 68-69: " ...subcellular localization, biological function, and their participation in hormone crosstalk and redox homeostasis" please complete with "of GASA proteins"

Response: Thanks for your suggestion. We changed the paragraph as suggested below.

  1. Line 66-82: here the authors tried to highlight the aim of the review, please rewrite it, to be more focused, as repetions are occuring, and cause diffuse sturture.

Response: Thanks for your suggestion. The paragraph has been revised.

  1. Please distinguish more clearly when you mention information about GASA proteins or GASA genes in all over the text. E.g:

In line 183:

"In Arabidopsis, AtGASA4 and AtGASA6 are located 183 at the cell periphery if they contain a signal peptide, however, they are found in the nu- 184 cleus if the signal peptide is missing [4]. In addition to the aforementioned subcellular distribution, AtGASA4 with non-cleavable signal peptide was speculated to attach to the endoplasmic reticulum (ER) [2]."

The genes are not indicated in italics, however in the reference studies genes were presented. Title of one of them:"Expression patterns of GASA genes in Arabidopsis thaliana: the GASA4 gene is up-regulated by gibberellins in meristematic regions"

Please try to make these parts more focused.

Or as 4. subsection is about protein localisation, remove all the information about genes, to the next section about "5. ...GASA genes".

Response: Thanks for your suggestion. We have corrected this remark as recommended. All information in the subsection 4 refers to GASA proteins.

  1. Again, it is not logical that in section 7. Modulation of GASA proteins’ expression: "The expression of GASA genes was determined in different studies. Overall, these studies showed that GASA genes are differentially expressed in different organs or tissues and under different treatments (Figure 2)." Why here again the localisation of genes are presented in a separeted section, and not involved in the section 5. Please try to follow a more logical structure and/or suitable subsection titles need.

Response: Thanks for your suggestion. Section 7 have been integrated into the other sections for a more focused arrangement.

  1. Figure 2 is not enough informative.

Response: Thanks for your suggestion. Figure 2 have been removed.

  1. From line 305 the relationship between GASA and hormones are mentioned, however the next section is about "Phytohormones and GASA proteins" So, please consider a focused arrangement.

Response: Thanks for your suggestion. We changed the section for a better structure.

Round 2

Reviewer 2 Report

The authors answered the questions, and performed a revision.